# Defective homologous recombination DNA repair as therapeutic target in advanced chordoma

Stefan Gröschel et al.[#]

Chordomas are rare bone tumors with few therapeutic options. Here we show, using whole-exome and genome sequencing within a precision oncology program, that advanced chordomas ($n = 11$) may be characterized by genomic patterns indicative of defective homologous recombination (HR) DNA repair and alterations affecting HR-related genes, including, for example, deletions and pathogenic germline variants of *BRCA2*, *NBN*, and *CHEK2*. A mutational signature associated with HR deficiency was significantly enriched in 72.7% of samples and co-occurred with genomic instability. The poly(ADP-ribose) polymerase (PARP) inhibitor olaparib, which is preferentially toxic to HR-incompetent cells, led to prolonged clinical benefit in a patient with refractory chordoma, and whole-genome analysis at progression revealed a *PARP1* p.T910A mutation predicted to disrupt the autoinhibitory PARP1 helical domain. These findings uncover a therapeutic opportunity in chordoma that warrants further exploration, and provide insight into the mechanisms underlying PARP inhibitor resistance.

Chordomas are rare tumors of the axial skeleton and skull base that arise from remnants of the embryonic notochord, a transient midline structure that guides vertebral development, provides patterning information for surrounding tissues, and ultimately regresses to form the nucleus pulposus in the intervertebral disc[1]. First-line treatment of chordoma is based on surgical resection and radiotherapy. However, due to the proximity of most chordomas to vital structures, especially at the skull base, local control is rarely achieved, resulting in a recurrence rate greater than 50%. Furthermore, locoregional or distant metastases occur in 30–40% of cases[2]. Systemic treatment of advanced disease is exceedingly difficult as chordomas are generally resistant to conventional chemotherapy, and no drugs are approved for this indication. Several targeted agents directed against PDGFRA/B, EGFR, or mTORC1 have yielded encouraging rates of disease stabilization, although objective responses are rare and the often slow growth rate of chordomas needs to be taken into account[2–4]. Blockade of brachyury, a notochordal transcription factor that drives chordoma development and is not expressed in most normal adult tissues[5], represents, in principle, a promising strategy to selectively target chordoma cells. However, transcription factors are notoriously difficult to inhibit with small molecules. Thus, there remains an urgent need for novel therapeutic strategies to improve clinical outcomes in chordoma patients.

Whether insights into the genomic landscape of sporadic chordoma might provide new entry points for targeted therapies remains incompletely understood. Earlier studies employing microarray technologies, fluorescence in situ hybridization, quantitative PCR, and targeted sequencing of select cancer genes showed that chordomas are primarily characterized by non-random DNA copy number losses across the genome, frequently involving CDKN2A, PTEN, and chromatin regulatory genes such as SMARCB1 as potentially actionable alterations, as well as recurrent gains of the TBXT gene encoding brachyury[6–9]. More recently, a survey of single-nucleotide variants (SNVs), small insertions/deletions (indels), structural rearrangements, and copy number changes using a combination of whole-exome sequencing (WES), whole-genome-sequencing (WGS), and targeted sequencing identified recurrent alterations in additional loci not previously implicated in chordoma, such as ARID1A, encoding a subunit of the SWI/SNF chromatin remodeling complex, and LYST, whose protein product regulates lysosomal trafficking[10].

In this study, we have used WES and WGS to inform clinical decision making in patients with advanced chordoma who have exhausted standard treatment options. We observed that advanced chordomas may frequently harbor molecular alterations associated with impaired DNA repair via homologous recombination (HR) as potentially actionable genetic vulnerabilities. These results prompted experimental treatment with a poly (ADP-ribose) polymerase (PARP) inhibitor in a patient whose tumor was refractory to irradiation and medical therapy, which led to a prolonged response and enabled the discovery of mutational destabilization of the autoinhibitory PARP1 alpha-helical domain (HD) as an yet unrecognized mechanism underlying acquired PARP inhibitor resistance.

## Results

**WES and WGS of chordoma within a precision oncology program.** To identify therapeutically tractable molecular lesions, we performed WES ($n = 9$) and WGS ($n = 2$) of tumor tissue and matched blood from 11 patients (age, 27–72 years) with locally advanced and/or metastatic chordoma who were enrolled in the MASTER (Molecularly Aided Stratification for Tumor Eradication Research) program, a registry trial for younger adults with

advanced cancer across all histologies and patients with rare tumors[11]. All patients had previously received radiotherapy to the primary tumor site, following surgical resection in nine of 11 cases. Systemic treatment had been administered in six of 11 cases (imatinib, $n = 4$; imatinib/sirolimus, $n = 1$; imatinib followed by sunitinib, imatinib/everolimus, and erlotinib/bevacizumab, $n = 1$; doxorubicin and ifosfamide, $n = 1$). All patients had progressive disease prior to molecular analysis. Detailed clinical information is provided in Supplementary Table 1. Sequencing parameters are provided in Supplementary Tables 2 and 3. Consistent with a recent analysis of the genomic landscape of sporadic chordoma[10], tumors had a modest burden of non-synonymous somatic mutations (Supplementary Data 1), ranking them among cancers such as prostate adenocarcinoma and neuroblastoma[12]. In contrast, analysis of DNA copy number profiles showed high numbers of structural variants greater than 10 million base pairs (mbp) in size in the majority of cases (Fig. 1a, b and Supplementary Fig. 1). Recurrent deletions of chromosome 9p21.3 encompassing the cell cycle regulatory genes CDKN2A/B were found in all tumors, corroborating previous karyotypic and molecular cytogenetic findings that led to the notion that chordomas might be amenable to CDK4/6 inhibition[13], a hypothesis that is being explored in a phase 2 clinical trial (ClinicalTrials.gov Identifier NCT03110744).

**Alterations of HR DNA repair genes in chordoma.** Given that structural rearrangements may be caused by defective repair of DNA double-strand breaks via HR[14], we explored the possibility that HR deficiency contributes to chordoma development. We first compiled a list of 23 candidate HR genes. Specifically, we selected 12 genes that were assessed as a biomarker to stratify patients with castration-resistant prostate cancer for olaparib treatment in a recent phase 2 clinical trial and 11 additional genes reported to be involved in DNA damage repair or sensitivity to PARP inhibition (Supplementary Table 4)[15]. Examination of individual loci revealed that two patients with no previous history of cancer had pathogenic (Class 5 according to the American College of Medical Genetics and Genomics [ACMG] classification system; https://www.ncbi.nlm.nih.gov/clinvar) germline variants in established DNA repair genes (Supplementary Data 1). A heterozygous germline BRCA2 frameshift mutation (p.T3085fs*26) was accompanied by somatic deletion of the wild-type allele and co-occurred with biallelic somatic PTEN alterations (p.R233X mutation and loss of heterozygosity) in patient Chord_03. A heterozygous germline NBN frameshift mutation (p.K219Nfs*16) was accompanied by somatic deletion of the wild-type allele in patient Chord_06. In addition, patient Chord_01 had a heterozygous germline CHEK2 missense variant (p.R145W) that leads to destabilization of CHEK2 and failure of the S-phase checkpoint and has been classified as likely pathogenic (ACMG Class 4)[16]. This variant was accompanied by somatic deletion of the CHEK2 wild-type allele and co-occurred with biallelic somatic PTEN alterations (heterozygous p.G251V mutation and deletion of the wild-type allele). Three patients had germline variants of uncertain significance (ACMG Class 3) in FANCG, RAD51B, and RPA1, respectively, whose allele frequencies were not increased in the tumor compared with the normal control sample.

**Genomic imprints of defective HR DNA repair in chordoma.** In addition to structural rearrangements and alterations of individual HR DNA repair genes, we detected seven known mutational signatures (http://cancer.sanger.ac.uk/cosmic/signatures)[17]. Signature Alexandrov-COSMIC 3 (AC3), associated with defective HR, was found in all samples, and the 95% confidence interval of

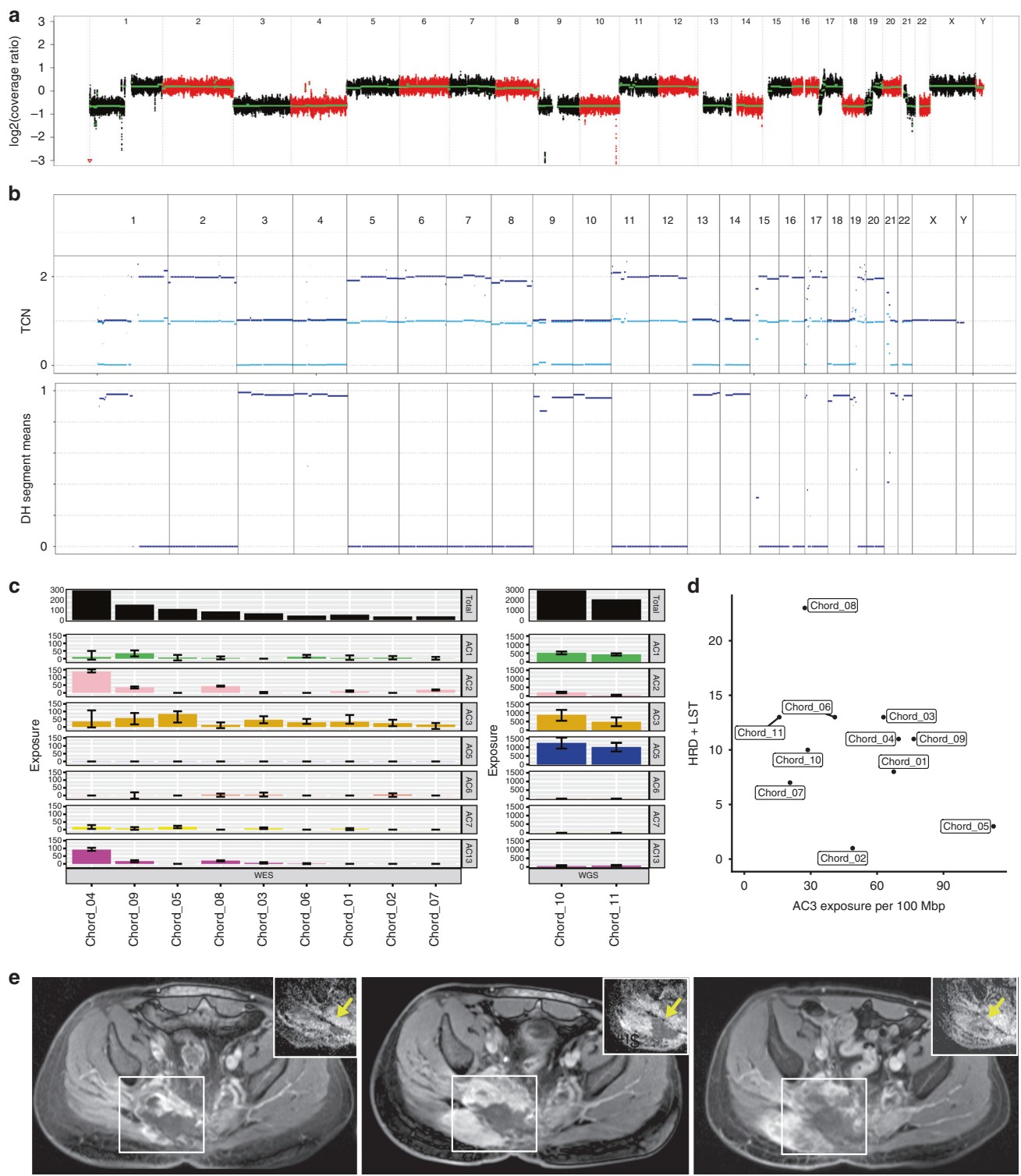

the exposure to AC3 excluded zero in eight of 11 samples (72.7%; Fig. 1c). Comparison of the signatures identified in our cohort to a background of 7042 cancer samples (WGS, $n = 507$; WES, $n = 6,535$)[17] showed significant enrichment of AC3 ($P = 4.52 \times 10^{-6}$). Signature AC3 co-occurred with extensive genomic instability, as illustrated by HR deficiency (HRD) scores and high numbers of large-scale state transitions (LSTs; Fig. 1d)[18].

**Actionability of defective HR DNA repair in chordoma.** Patient Chord_05, while not meeting traditional criteria for defective HR

such as biallelic inactivation of *BRCA1/2*, showed an exceptional exposure to the mutational signature AC3 and a high degree of genomic instability (Fig. 1a–d). This patient, a 57-year-old man, had been diagnosed with inoperable sacrococcygeal chordoma 80 months prior to enrollment in the MASTER program and underwent proton radiation therapy. Following progression 68 months after first diagnosis, the tumor proved refractory to repeat irradiation and, after initial stabilization, to systemic treatment with the tyrosine kinase inhibitor imatinib, as evidenced by rapid tumor growth and worsening of clinical

**Fig. 1** HR deficiency as clinically actionable feature in chordoma. **a** Copy number plot of patient Chord_05 showing chromosomal coordinates based on WES data (horizontal axis) and the log2 ratio of copy number changes (vertical axis). Red and black regions indicate different chromosomes. **b** CNA profile of patient Chord_05. Segment-wise total copy number counts after correction for TCC and ploidy are shown. **c** Contribution of mutational signatures (absolute exposures) to the overall SNV load in chordoma patients. Each bar represents the number of SNVs explained by the respective mutational signature in an individual tumor. Error bars represent 95% confidence intervals. Exposures for tumors analyzed by WES are displayed on the left. Exposures for tumors analyzed by WGS are displayed on the right. AC1 clock-like, spontaneous deamination; AC2 and AC13 altered APOBEC activity; AC3 defective HR; AC6 defective DNA mismatch repair; AC7 ultraviolet light exposure; AC10 altered POLE activity. **d** Scatter plot of measures of genomic instability (sum of HRD score and number of LSTs; vertical axis) versus exposures to signature AC3 (horizontal axis). To include both WES and WGS data, exposures to AC3 were normalized to the size of the target capture. **e** Therapeutic targeting of defective HR in patient Chord_05. T1-weighted, fat-saturated, post-contrast MRI at baseline 1 (left panel), after 6 months of imatinib therapy (progressive disease, baseline 2 for further follow-up; middle panel), and after 5 months of olaparib therapy (stable disease compared to baseline 2; right panel). A biopsy for WES was taken at progression (middle panel). The main bulk of the sacrococycgeal chordoma is located right to the midline with infiltration of the pelvis and the gluteal muscles (white rectangles). Corresponding apparent diffusion coefficient (ADC) maps derived from diffusion-weighted imaging of the tumor area are shown in the top right corner of each panel. Compared to baseline 2, a reduction of tumor bulk, especially the intrapelvic component, and increased necrosis, as indicated by new areas with lack of contrast enhancement, were seen. An increase in ADC from 1030 $mm^2s^{-1}$ to 1352 $mm^2s^{-1}$ between both time points indicates a reduction in cellularity (yellow arrows)

symptoms such as pain, incontinence, and walking disability. Given that defects in HR impair the repair of DNA double-strand breaks caused by chemotherapeutic agents such as cisplatin, and are synthetic lethal to inhibition of PARP[19], the tumor's genomic profile was considered actionable and off-label therapy with the PARP inhibitor olaparib at 800 mg daily was initiated. Besides mild neutropenia and anemia (Common Terminology Criteria for Adverse Events Grade 1–2), treatment was well tolerated, tumor symptoms gradually decreased after 2 months, and the patient was ambulatory again. Magnetic resonance imaging (MRI) after 5 months of treatment demonstrated that disease progression was halted, with the tumor showing signs of necrosis, partial volume shrinkage, and decreased cellularity (Fig. 1e).

**Olaparib resistance due to impaired PARP1 autoinhibition.** Regrettably, follow-up MRI of the pelvis after 10 months of olaparib treatment showed progressive disease (Fig. 2a), which was reflected by increased pain in the sacrum radiating to the perineal region and the thighs as well as recurrent incontinence. To search for genetic mechanisms underlying the acquired resistance to PARP inhibition, a repeat biopsy from a progressive region of the tumor (Chord_05R) was subjected to WGS. This analysis revealed a highly rearranged DNA copy number profile with multiple structural and numerical changes distributed across the genome, including highly complex structural rearrangements of chromosomes 6 and 15 (Supplementary Fig. 2). In addition, we identified an increase in the number of coding non-synonymous mutations (SNVs, 60 versus 40; indels, 14 versus 4) compared with the exome sequence obtained before olaparib treatment (Supplementary Data 1). Among the newly gained variants were a *TP53* frameshift mutation (A83fs; mutant allele frequency, 55%) and a *PARP1* missense variant (p.T910A; mutant allele frequency, 28%) whose presence was confirmed by Sanger sequencing (Fig. 2b). Analysis of the variant allele fractions from the two samples using a look-up procedure demonstrated that the variants private to either specimen outnumbered those shared by both, indicating that the separation of the clones active before and after olaparib treatment occurred early in the growth history of this tumor (Supplementary Fig. 3).

To investigate whether the *PARP1* p.T910A mutation provided a mechanistic explanation for the secondary failure of olaparib treatment, thereby validating PARP1 as a therapeutic target in this tumor, we performed protein structure modeling analyses. Threonine 910 is located in a loop region adjacent to the enzymatic site that is part of the interface between the signature ADP-ribosyl transferase (ART) fold and the regulatory HD of the PARP1 catalytic domain (Fig. 2c, d). However, similar to

mutations of residues Y848 and A925 in the ART domain that were recently identified in a CRISPR-based mutagenesis screen for PARP inhibitor resistance alleles[20], the p.T910A variant does not appear to have a direct effect on olaparib binding as T910 is not part of the $NAD^+$ binding pocket and does not contact the drug. In addition to binding the catalytic site, thereby blocking $NAD^+$ access and poly(ADP-ribose) formation, olaparib is thought to inhibit PARP1 function through at least two additional mechanisms. First, it traps PARP1 on DNA single-strand breaks via conformational changes in the active site loop, resulting in obstructed replication forks that require HR to be resolved[20–22]. Second, olaparib favors the inactive conformation of PARP1 by stabilizing the HD, which is critical for allosteric autoinhibitory interactions as illustrated by the constitutive activity of a PARP1 mutant lacking the HD, even in the presence of olaparib bound to the catalytic pocket[23,24]. To gain insight into the functional consequences of the p.T910A variant, we performed energy calculations[25], which predict the stabilizing or destabilizing effect of a mutation as negative or positive difference in free energy (ddG) between the mutant and wild-type proteins. The p.T910A variant was overall predicted to destabilize the structure of the PARP1 catalytic domain (Supplementary Table 5). However, calculations employing a constitutively active PARP1 structure lacking the autoinhibitory HD[23] indicated a weak destabilizing effect (ddG = 0.233 Rosetta Energy Units [REU]; Fig. 2e), whereas predictions on the ART domain and the HD of the PARP1-DNA complex structure yielded a much stronger destabilizing effect (ddG = 1.865 REU; Fig. 2d), possibly due to different local packing of the T910 side chain and/or propagation of the perturbation to the interaction interface with the HD. This indicates that the p.T910A mutation acts allosterically to destabilize HD-mediated autoinhibition, favoring an active-like conformation that overrides PARP1 inhibition by olaparib. In line with experimental data obtained in the context of HD-deficient PARP1[23], our simulations suggest that this effect will be independent of the presence of olaparib as there was only a slight difference in free energy (ddG = 0.233 REU; Fig. 2e) following binding of the inhibitor.

**Discussion**

Our data suggest that advanced and extensively pretreated chordomas are recurrently characterized by genomic alterations associated with defective DNA repair via the HR pathway, including biallelic germline and somatic mutations of individual genes involved in this process, enrichment of a specific mutational signature induced by HR deficiency, and increased genomic instability. Interestingly, only three of 11 patients carried genetic alterations

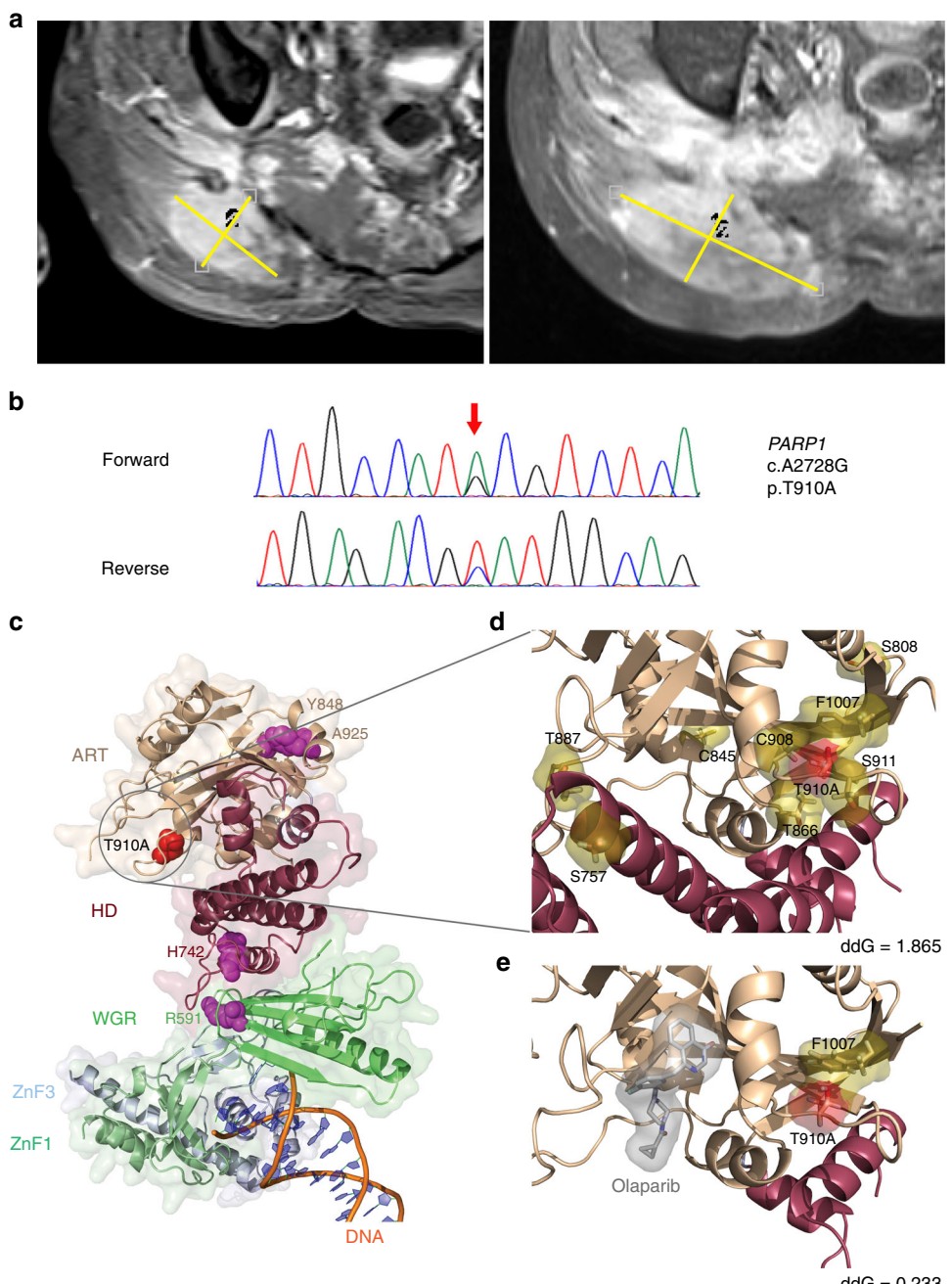

**Fig. 2** Acquired resistance to olaparib treatment in chordoma. **a** T1-weighted, fat-saturated, post-contrast MRI after 7 months (stable disease; left panel) and 10 months (progressive disease; right panel) of olaparib therapy. A biopsy for WGS was taken at progression (right panel). **b** Missense mutation in *PARP1* exon 20 detected by Sanger sequencing. Nucleotide (arrow) and amino acid substitutions are given next to the chromatograms. Sequence numbering is according to NCBI Reference Sequences NM_001618 and NP_001609. **c** Structure of p.T910A-mutant PARP1 bound to DNA (PDB ID: 4DQY). Side chains of amino acids whose mutation has been linked to PARP1 inhibitor resistance (T910, this study; R591, H742, Y848, and A925, ref. [19]) are represented as spheres. WGR tryptophan-glycine-arginine-rich domain; ZnF1 zinc-finger domain 1; ZnF3 zinc-finger domain 3. **d** Detail view of the p.T910A mutation site in the PARP1-DNA complex structure (PDB ID: 4DQY). Threonine 910 and amino acids whose side chains are displaced according to energy calculations (cut-off, 0.05 Å) are represented as red and olive sticks and surfaces, respectively. **e** Detail view of the p.T910A mutation site in the structure of constitutively active PARP1 (PDB ID: 5DS3) in complex with olaparib (represented as gray sticks and surface)

that would meet traditional criteria for HR deficiency (biallelic inactivation of *BRCA2*, *NBN*, and *CHEK2*, respectively), which highlights the need to develop novel tools for identifying tumors with functional defects similar to those associated with BRCA1/2 inactivation. The potential of compound genomic measures in this regard is illustrated by our findings in patient Chord_05 who showed a striking exposure to signature AC3 and a high degree of

genomic instability, as evidenced by an elevated HRD score and a high number of LSTs. Although the underlying mutations were not identified unambiguously, these features by themselves formed the basis for successful therapy with single-agent olaparib, and acquired resistance was associated with a newly gained PARP1 mutation that is predicted to restore enzymatic activity in the presence of drug, thereby validating PARP1 as therapeutic target in this tumor. Of

note, most patients in our study also exhibited monoallelic HR gene deletions. However, the question of whether heterozygous loss of multiple HR genes in the context of structural rearrangements leads to, or is reflective of, HR deficiency, and how pathogenic variants in other genes may contribute to chordoma development cannot be answered by our study.

The observed prevalence of an HR deficiency footprint in advanced chordomas, which is reminiscent of the genomic features of breast, ovarian, and prostate cancer in which HR deficiency has emerged as a therapeutic liability[15,19], provides a rationale for genomics-guided therapy using agents, either alone or in combination, that are preferentially toxic to HR-incompetent cells, such as PARP inhibitors, platinum derivatives, or trabectedin. Furthermore, given recent preclinical reports on the efficacy of immune checkpoint blockade in HR-deficient tumors, the results presented herein and the finding of PD-1 and PD-L1 expression in the chordoma microenvironment may warrant clinical trials of immune checkpoint inhibitors in chordoma[26,27]. Of note, all tumors had been exposed to radiotherapy prior to molecular analysis, raising the interesting possibility that this treatment may have shaped their genomic profiles and associated therapeutic vulnerabilities. Therefore, even though the imprint of ionizing radiation in cancer is not well understood[28] and previous microarray analyses that included radiation-naïve cases indicate that multiple DNA copy number alterations are an intrinsic feature of chordoma[8], future studies should address whether radiotherapy or other extrinsic conditions, such as exposure to cytotoxic drugs, might result in conditional synthetic cytotoxicity that can be exploited therapeutically[29].

In addition to identifying PARP inhibition as a molecularly guided strategy to target refractory chordomas, our data also provide insight into the mechanisms underlying acquired resistance to this treatment modality. While previous in vitro studies uncovered that the selective pressure provided by PARP inhibitors can lead to restoration of HR via revertant mutations in BRCA1/2 or RAD51C/D, inactivation of 53BP1 or REV7, and loss of PARP1 expression[30], the importance of PARP1 mutations has only recently emerged based on forward genetic screens and the observation of a p.R591C variant in the PARP1 tryptophan-glycine-arginine-rich domain in an ovarian cancer patient who showed de novo resistance to olaparib[20]. We now expand these data by reporting the first example of a secondary mutation, p.T910A, in the PARP1 HD that occurred in a patient whose disease progressed after 10 months of olaparib treatment, and protein structure modeling analyses supported a mechanism of olaparib resistance in which the p.T910A allele restores PARP1 activity in the presence of the drug by disabling HD-mediated autoinhibition. From a therapeutic perspective, it will be particularly interesting to investigate whether other PARP inhibitors currently tested in clinical trials, such as rucaparib and talazoparib, may overcome p.T910A-mediated resistance as our analysis of the mutation's impact on the ART domain-HD interface demonstrated that these drugs have additional contacts with residues in the HD (e.g., Q759, E763, and D766) when compared to olaparib (Supplementary Fig. 4), which might increase their capacity to stabilize the autoinhibited conformation.

In summary, our study has uncovered a biological feature of advanced chordoma that represents an immediately actionable therapeutic target and provides a rationale for genomics-guided clinical trials of pharmacologic PARP inhibition in this intractable tumor entity. More broadly, our findings illustrate that the concept of "BRCAness" as clinically actionable genomic feature extends beyond common epithelial cancers, and further our understanding of the mechanisms underlying acquired resistance to this important class of targeted cancer therapeutics.

## Methods

**Patient samples.** For WES and WGS, fresh-frozen tumor and matched germline control specimens were obtained from adult patients diagnosed with chordoma according to World Health Organization criteria at four German cancer centers (NCT Heidelberg and Heidelberg University Hospital, Heidelberg; West German Cancer Center, Essen; University Hospital Carl Gustav Carus, Dresden; Frankfurt University Hospital, Frankfurt). Disease manifestations and previous treatments are given in Supplementary Table 1. Prior to processing, samples were pseudonymized, and histology and cellularity of the tumors were determined at the Institute of Pathology, Heidelberg University Hospital. All patients provided written informed consent under a protocol approved by the Ethics Committee of Heidelberg University, and the study was conducted in accordance with the Declaration of Helsinki.

**Isolation of analytes.** DNA from tumor and control specimens was extracted at the DKFZ-HIPO Sample Processing Laboratory using the AllPrep DNA/RNA/Protein Mini Kit (Qiagen). DNA quality was assessed with a 2100 Bioanalyzer system (Agilent), and DNA quantification was performed using a Qubit 2.0 Fluorometer (Invitrogen).

**WES.** SureSelect Human All Exon in-solution capture reagents (Agilent) were used for exome capturing. The specific versions of the target captures used for the different samples are given in Supplementary Table 2. Briefly, 1.5 μg genomic DNA was fragmented to 150–200 base pairs (bp) insert size with a Covaris S2 device, and 250 ng of Illumina adapter-containing libraries were hybridized with exome baits at 65 °C for 16 h. Paired-end sequencing (2 × 101 bp) was carried out with a HiSeq 2500 instrument (Illumina) for samples Chord_01, Chord_02, Chord_03, Chord_04, and Chord_07 and with a HiSeq 4000 instrument (Illumina) for samples Chord_05, Chord_06, Chord_08 and Chord_09.

**WGS.** Library preparation was performed with the TruSeq Nano Library Preparation Kit (Illumina). A HiSeq X instrument (Illumina) was used for paired-end sequencing (2 × 151 bp).

**Mapping of sequencing data.** Read mapping was performed using the 1000 Genomes Phase 2 assembly of the Genome Reference Consortium human genome (build 37, version hs37d5) concatenated with the genome of Enterobacteria phage phiX174 using BWA mem (version 0.7.15; parameter -T 0 and all other parameters set to default). BAM files were sorted with bamsort (part of the package biobambam, version 0.0.148), and duplicates were marked with markdup (part of the package Sambamba, version 0.6.5)[31]. Sequencing coverage and quality scores can be found in Supplementary Table 2.

**Detection of SNVs and small indels.** An in-house computational analysis pipeline derived from SAMtools mpileup and bcftools was used for detection of somatic SNVs from paired tumor and control specimens. Adjustments of parameters and heuristic filtering were performed as described previously[32–34]. For determining germline and somatic variants, a pileup of the bases in the paired control specimen was computed for each SNV position by SAMtools mpileup with parameters -Q 0 -q 1. After annotation with GENCODE (release 19) using ANNOVAR (version November 2014), somatic non-synonymous coding variants of high confidence were selected. The analysis of mutational signatures was based on all high-confidence somatic variants. Small indels were identified by Platypus (version 0.8.1; parameters ploidy = 2, nIndividuals = 2) by providing matched tumor and control BAM files[35]. To be considered as high-confidence, somatic calls (control genotype 0/0) were required to either have the Platypus filter flag PASS or pass custom filters allowing for low variant frequency using a scoring scheme. Indels were annotated with ANNOVAR (version February 2016), and somatic high-confidence indels falling into a coding sequence or splice site were extracted.

**Supervised analysis of mutational signatures.** For linear combination decomposition of individual mutational catalogs with known mutational signatures (http://cancer.sanger.ac.uk/cosmic/signatures), non-negative least squares (NNLS) were computed using the R package YAPSA (Yet Another Package for Signature Analysis)[36,37]. Mutational catalog correction was performed to account for differences in the occurrence of triplet motifs by comparing the whole genome to WES capture regions (function normalizeMotifs_otherRownames in YAPSA). In order to improve specificity, the NNLS computation was performed twice. Only signatures with exposures, i.e., contributions in the linear combination, reaching a certain cut-off were kept after the first execution, and the NNLS algorithm was repeated with the reduced set of signatures. Since the detection of signatures may differ, a random operator characteristic analysis was performed to determine signature-specific cut-offs using the mutational catalogs of 7042 tumor samples (WGS, $n = 507$; WES, $n = 6,535$)[17] and mutational signatures from COSMIC: AC1: 0; AC2: 0.0104594; AC3: 0.0819406; AC4: 0.0175397; AC5: 0; AC6: 0.0015485; AC7: 0.040133; AC8: 0.242755; AC9: 0.1151714; AC10: 0.0100838; AC11: 0.0992488; AC12: 0.2106201; AC13: 0.0078766; AC14: 0.1443059; AC15: 0.0379603; AC16: 0.3674349; AC17: 0.002648; AC18: 0.3325386; AC19: 0.1156454;

AC20: 0.1235028; AC21: 0.1640255; AC22: 0.0310222; AC23: 0.0333866; AC24: 0.0324018; AC25: 0.0161191; AC26: 0.0933522; AC27: 0.0093201; AC28: 0.0561643; AC29: 0.0593621; AC30: 0.0591536. The specific cut-offs can also be retrieved from YAPSA with the following R code: library(YAPSA), data(cut-offs), cutoffCosmicValid_rel_df[6,]. Confidence intervals were determined by using the concept of profile likelihoods[38]. Likelihoods were assessed from the residue distribution after NNLS decomposition (initial model of the data). The confidence interval of a given signature was determined as follows: the exposure to this signature was perturbed and fixed as compared to the initial model. The exposures to remaining signatures were again computed using the NNLS algorithm, resulting in an alternative model with one degree of freedom less. Likelihoods were then computed from the distribution of residuals of the alternative model. Next, a likelihood ratio test for the log-likelihoods of the initial and alternative models was performed, yielding a test statistic and a $P$ value for the perturbation. To determine the limits of 95% confidence intervals, the Gauss-Newton method-based R package pracma was used for computing perturbations corresponding to $P$ values of $0.05/2 = 0.025$ (two-sided). Mutational signatures derived from chordoma cases were compared to mutational signatures derived from a background of cancer specimens (WGS, $n = 507$; WES, $n = 6,535$)[17] by Fisher exact tests and subsequent multiple testing correction using the Benjamini-Hochberg method.

**Look-up analysis of variant allele fractions**. The sets of mutations detected in patient Chord_05 before and after olaparib treatment were merged, and a look-up of read counts for the reference and alternative alleles was performed with Platypus (version 0.8.1.1; option callVariants). All variants with a minimum coverage of 20 in both samples were kept for further analysis. The output of Platypus was read into R and parsed for the read counts, germline mutations were removed, and variant allele fractions were displayed in a scatter plot.

**Analysis of copy numbers, tumor cell content, and ploidy**. ACEseq[39] was used for calling allele-specific copy number alterations (CNAs) from WGS data. Absolute allele-specific copy numbers, tumor cell content (TCC), and ploidy were determined by computing coverage ratios of tumor and control as well as the B allele frequency (BAF) of heterozygous single-nucleotide polymorphisms (SNPs). Structural variants called with SOPHIA were included to enable better genome segmentation. To prevent biases due to oversegmentation, copy number profiles were further smoothed prior to calculating the total number of gains and losses. Segments smaller than 3 Mbp were merged with the neighboring segment to which they had the smallest difference between respective total copy number values. Based on the resulting segments, the number of gains and losses was estimated.

CNAs were inferred from WES data with cnvKit (version 0.9.3) using default parameters. SNPs were determined as heterozygous if the alternative allele fraction ranged between 0.3 and 0.7 in the paired normal control. Segments covering at least 20 heterozygous SNPs were used to infer TCC, ploidy, and allele-specific copy number estimates. Segments were categorized as balanced or imbalanced based on the distribution of alternative SNP allele frequencies. A segment was classified as balanced if the global maximum of the distribution ranged between 0.45 and 0.55. Remaining segments were categorized into two groups, ambiguous segments containing one density peak outside the 0.45–0.55 interval, and imbalanced segments containing two peaks. Ambiguous segments were subsequently neglected. The mean BAF of all SNPs in the segment that were heterozygous in the germline was calculated for imbalanced segments using the allele with the higher read count as B allele. The mean B allele read count was computed as the product of the total coverage and the BAF of the respective segment.

An adapted ACEseq method was used for TCC and ploidy estimation of a sample. To estimate TCC, values of 0.15–1.0 were tested, and a range of 1.0–6.5 was allowed for estimation of ploidy. A segment-wise estimation[40] of absolute and allele-specific copy numbers as well as the decrease in heterozygosity (DH) were computed for every possible TCC and ploidy combination. For balanced segments, allele-specific copy numbers were computed as total copy number divided by two. For imbalanced segments, a function of coverage and B allele read counts was applied. For calculation of total and allele-specific copy numbers, the weighted mean distance of all segments to the next integer copy number state was computed, where allowed means even total copy number states for balanced segments and any integer copy number state for imbalanced segments and allele-specific copy numbers. TCC/ploidy combinations with negative copy number states or a DH above one for any segment were excluded. Local minima in the weighted mean distance were considered as possible TCC/ploidy solution for the sample and were visually inspected. In addition, TCC was determined from the mutant allele fraction distribution for somatic SNVs, and CNA- and SNV-based estimates were compared. For sample Chord_07, TCC was manually adjusted to 20%.

**HR deficiency and LST scores**. For estimating stable HRD and LST scores, oversegmentation caused by technical noise was reduced by smoothing of copy number profiles. Segments smaller than 3 Mbp were merged with their more similar neighbor as previously described[18]. Any switch between copy number states of segments larger than 10 Mbp that did not correspond to entire chromosome arms was counted as LST[18]. Additionally, subchromosomal segments larger than

15 Mbp and corresponding to loss of heterozygosity were counted for the HRD estimation.

**Energy calculations**. Calculations were carried out considering both the constitutively active PARP1 structure (Protein Data Bank Identifier [PDB ID]: 5DS3) and the region corresponding to the ART domain and the HD (i.e. amino acids 662–1011) of the PARP1-DNA complex structure (PDB ID: 4DQY)[23,41]. Structures were pre-minimized to solve potential clashes using the Rosetta Relax protocol[42], applying backbone restraints to avoid excessive displacements from the starting X-ray structures. Fifty energy calculation cycles were performed in low-resolution (i.e., with a fixed backbone) and high-resolution (i.e., allowing backbone degrees of freedom) modes using the Rosetta ddG_monomer protocol[25], and ddG values are expressed as REU. Since the same positive ddG trend was observed in all simulations, results are shown only for the low-resolution mode, which gave the highest ddG values. To detect amino acids perturbed by the p.T910A mutation, side chain conformations were compared between the top-scoring mutant and wild-type conformations generated during the low-resolution simulations, and amino acids with at least one side-chain atom displacement greater than 0.05 Å were selected employing ad hoc created biopython scripts[43]. Structure representations were drawn with PyMOL 1.7.x (https://pymol.org).

**Reporting summary**. Further information on experimental design is available in the Nature Research Reporting Summary linked to this article.

## Data availability
Sequencing data were deposited in the European Genome-phenome Archive under accession code EGAS00001002720 [https://www.ebi.ac.uk/ega/studies/EGAS00001002720].

## Code availability
All code used is available upon request.

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

## Acknowledgements

The authors thank the DKFZ-HIPO Sample Processing Laboratory and the DKFZ Genomics and Proteomics Core Facility for technical support. We also thank K. Beck, K. Willmund, and P. Lichter for infrastructure and program development within DKFZ-HIPO. Tissue samples were provided by the NCT Heidelberg Tissue Bank in accordance with its regulations and after approval by the Ethics Committee of Heidelberg University. This work was supported by grant H021 from DKFZ-HIPO. S.G. is the recipient of a Starting Grant (677209) from the European Research Council. D.H. is a member of the Hartmut Hoffmann-Berling International Graduate School of Molecular and Cellular Biology and of the MD/PhD Program of Heidelberg University. F.R. is supported by a Humboldt Research Fellowship and a grant from the Michael J. Fox Foundation. F.R. and R.B.R. are supported by the Cluster of Excellence "Cellular Networks" funded by the German Federal and State Governments and implemented by the German Research Foundation and the German Council of Science and Humanities.

## Author Contributions

S.G., D.H., G.W., M.F., B.H., K.K., S.U., M.S. and S.F. analyzed and interpreted bioinformatics data. F.R. and R.B.R. performed protein structure modeling analyses. S.G., P.H., O.M., S.K., C.H., C.E.H., C.B., S.R., S.B. and S.F. contributed patient samples. B.K., L.G. and E.S. interpreted germline variants. D.B. analyzed and interpreted imaging data. W.W., R.P. and A.S. performed pathology review. P.C., J.M., M.G., D.R., E.R., K.P., R.E., S.W., C.v.K., C.S., R.F.S., R.B.R. and B.B. provided essential reagents, expertise, and infrastructure. S.G., D.H. and S.F. wrote the manuscript, which was reviewed and edited by all co-authors. S.G., D.H., H.G., M.S. and S.F. conceived and supervised the project.

## Additional information

**Competing interests:** The authors declare no competing interests.

Stefan Gröschel[1,2,3], Daniel Hübschmann[4,5,6,7], Francesco Raimondi[8,9], Peter Horak[3,10], Gregor Warsow[4,11], Martina Fröhlich[3,12], Barbara Klink[13,14], Laura Gieldon[13,14], Barbara Hutter[3,12], Kortine Kleinheinz[4,15], David Bonekamp[16], Oliver Marschal[17], Priya Chudasama[3,10], Jagoda Mika[1,18], Marie Groth[10,18], Sebastian Uhrig[3,12,18], Stephen Krämer[4,18], Christoph Heining[14,19], Christoph E. Heilig[10], Daniela Richter[14,19], Eva Reisinger[4,11], Katrin Pfütze[3,20], Roland Eils[3,4,20], Stephan Wolf[3,21], Christof von Kalle[3,20,22],

Christian Brandts[23,24], Claudia Scholl[3,25], Wilko Weichert[26,27], Stephan Richter[14,28], Sebastian Bauer[29,30], Roland Penzel[3,31], Evelin Schröck[13,14], Albrecht Stenzinger[3,31], Richard F. Schlenk[2,3,32,33], Benedikt Brors [3,12], Robert B. Russell [8,9], Hanno Glimm[14,19], Matthias Schlesner [3,4,34] & Stefan Fröhling[3,10,20]

[1]Molecular Leukemogenesis Group, German Cancer Research Center (DKFZ), 69120 Heidelberg, Germany. [2]Department of Internal Medicine V, Heidelberg University Hospital, 69120 Heidelberg, Germany. [3]German Cancer Consortium (DKTK), 69120 Heidelberg, Germany. [4]Division of Theoretical Bioinformatics, DKFZ, 69120 Heidelberg, Germany. [5]Division of Stem Cells and Cancer, DKFZ, 69120 Heidelberg, Germany. [6]Heidelberg Institute for Stem Cell Technology and Experimental Medicine, 69120 Heidelberg, Germany. [7]Department of Pediatric Immunology, Hematology and Oncology, Heidelberg University Hospital, 69120 Heidelberg, Germany. [8]BioQuant, Heidelberg University, 69120 Heidelberg, Germany. [9]Heidelberg University Biochemistry Center, 69120 Heidelberg, Germany. [10]Division of Translational Medical Oncology, National Center for Tumor Diseases (NCT) Heidelberg and DKFZ, 69120 Heidelberg, Germany. [11]Omics IT and Data Management Core Facility, DKFZ, 69120 Heidelberg, Germany. [12]Division of Applied Bioinformatics, DKFZ and NCT Heidelberg, 69120 Heidelberg, Germany. [13]Institute for Clinical Genetics, Faculty of Medicine Carl Gustav Carus, Technische Universität Dresden, 01307 Dresden, Germany. [14]DKTK, 01307 Dresden, Germany. [15]Department for Bioinformatics and Functional Genomics, Institute for Pharmacy and Molecular Biotechnology and BioQuant, Heidelberg University, 69120 Heidelberg, Germany. [16]Division of Radiology, DKFZ, 69120 Heidelberg, Germany. [17]Onkologische Schwerpunktpraxis, 38100 Braunschweig, Germany. [18]Faculty of Biosciences, Heidelberg University, 69120 Heidelberg, Germany. [19]Department of Translational Medical Oncology, NCT Dresden and University Hospital Carl Gustav Carus, 01307 Dresden, and DKFZ, 69120 Heidelberg, Germany. [20]DKFZ-Heidelberg Center for Personalized Oncology (HIPO), 69120 Heidelberg, Germany. [21]Genomics and Proteomics Core Facility, DKFZ, 69120 Heidelberg, Germany. [22]Division of Translational Oncology, National Center for Tumor Diseases (NCT) Heidelberg and DKFZ, 69120 Heidelberg, Germany. [23]University Cancer Center Frankfurt (UCT), Department of Medicine, Hematology/Oncology, Goethe University, 60595 Frankfurt, Germany. [24]DKTK, 60595 Frankfurt, Germany. [25]Division of Applied Functional Genomics, DKFZ, 69120 Heidelberg, Germany. [26]Institute of Pathology, Technical University Munich, 81675 Munich, Germany. [27]DKTK, 81675 Munich, Germany. [28]Department of Internal Medicine I, University Hospital Carl Gustav Carus, 01307 Dresden, Germany. [29]West German Cancer Center, University of Duisburg-Essen, 45147 Essen, Germany. [30]DKTK, 45147 Essen, Germany. [31]Institute of Pathology, Heidelberg University Hospital, 69120 Heidelberg, Germany. [32]Department of Internal Medicine VI, Heidelberg University Hospital, 69120 Heidelberg, Germany. [33]NCT Trial Center, NCT Heidelberg and DKFZ, 69120 Heidelberg, Germany. [34]Bioinformatics and Omics Data Analytics Group, DKFZ, 69120 Heidelberg, Germany. These authors contributed equally: Stefan Gröschel, Daniel Hübschmann, Hanno Glimm, Matthias Schlesner, Stefan Fröhling.

