## [Peer Review File · Nature Communications]

Reviewers' comments:

Reviewer #1 (Remarks to the Author):

This is an interesting paper describing possible homologous recombination DNA repair defects in chordoma, reporting an interesting case with acquired PARP inhibitor resistance and a PARP1 mutation.

Major points:

1) Heterozygous deletion of ATR, BRCA2, CHEK2, ERCC6, FANCA, FANCC, FANCD2, FANCG, PALB2, RAD18, RAD51, RAD54L, or XRCC3 will not lead to homologous recombination defect. AC3 signature or HRD score may not be an absolute marker of homologous recombination defect. The role of PTEN in homologous recombination is controversial (Fraser M, et al. Clin Cancer Res. 2012).

Therefore, I am not convinced that the chordoma cases other than Chord_03 and Chord_06 are homologous recombination defective cases.

The sentence in the abstract "heavily pretreated chordomas (n=7) are invariably characterized by alterations affecting the homologous recombination (HR) DNA repair pathway..." is an overinterpretation and should be rewritten.

Similarly, the first sentence in Discussion "Our data indicate that a substantial portion of advanced-stage chordomas harbor molecular alterations that affect the repair of DNA damage via HR..." is also incorrect and should be rewritten.

Minor points:

1) Supplemental Table 1. Age and gender of the patients should be included.

2) For cases with germline mutations in BRCA2 or NBN, presence or absence of other tumors should be described.

Reviewer #2 (Remarks to the Author):

The authors report on WES on 7 normal tumor pairs from patients with pretreated advanced chordoma and report a stabilization of disease in a patient treated with olaparib. These findings are novel and the identification of HR deficiency as a potential target in this rare cancer type is of great interest. Furthermore, the development of a somatic PARP1 mutation in the patient treated with olaparib is quite interesting. However, some additional information is needed to interpret these results and the presentation of data seems overly simplistic. Are HR genes significantly over-represented relative to alterations in other genes? No formal analysis is presented to that effect. The authors include a whole list of genes, not all of which have clear impact on HR function in cancers. For example, the role of PTEN and some other genes on the list have not been clearly shown to drive an HRD phenotype either functionally or by mutational signature profiling. The results and discussion should be more nuanced in this regard. What is the difference between deletions and LOH. Are the deletions meant to be homozygous deletions. If not, then how do they differentiate deletions from LOH at that locus? The many deletions in HR gene if only heterozygous and not homozygous are of unclear importance, particularly if they are just passengers in larger chromosomal deletions. Therefore, reporting these "deletions" may be confusing to the reader unless they are occurring at significantly higher frequency in HR genes compared to genes in other functional pathways.

The driver of the chord_05 "brcaness signature" is unclear, but the authors imply that the exceptionally high AC3 signature is also related to the 13 HR gene alterations. A more likely explanation is the frequent LOH or heterozygous deletion of these 13 HR genes is a result of HRD not a driver of HRD, unless one of these events is biallelic. The authors should be clear that the signature of HRD is more important here in choosing a therapy than the many monoallelic deletions of putatively important HR genes. Unless these monoallelic deletions are over-represented

in HR genes relative to other genes, their inclusion in Figure 1, panel C is misleading. I don't think the use of the term BRCAness is helpful without a clear definition of what that means. For example, in the discussion, they reference the "high prevalence of BRCAness in heavily treated chordomas..." What do the authors mean by "BRCAness". This is a poorly defined term. They would be better off describing that 71% had a genomic mutational signature consistent with HRD or a clear biallelic mutation in an HR gene. As stated it is vague and confusing. The authors report methods for somatic variant calling but no methods for germline analyses. Furthermore, Figure 1 which summarizes the mutation data is not well annotated in the legend. "Recurrent somatic mutations" implies the same mutation is seen across cases. I think the authors mean to say recurrent somatically mutated "genes"

Reviewer #3 (Remarks to the Author):

The goal of this study was to identify genomic alterations that can be directly linked to targeted therapy in a rare cancer that shows high frequency of local recurrence and does not have too many treatment options other than surgery. The manuscript was nicely written. However, the study is really small (based on 7 WES) and there are some major problems in study design, data interpretation and conclusion.

1. With 7 WES, there is no power to identify driver events in these tumors.
2. The major conclusion of the impaired HR pathway is not convincing. Most of the alterations are copy number variants (CNVs), however, as shown in the CNV profile figures, most of these CNVs involved large genomic segments containing a huge number of genes. In fact, I did not see any of the listed HR genes showing focal deletions/amplifications.
3. In most cases, genes in a single pathways display mutations in a mutually exclusive manner, rather than co-occurrence, which usually reflects non-significant mutations.
4. All these patients were treated with either RT or chemo before tumor specimens were obtained. The genomic profile may be related to treatment.

RE: NCOMMS-18-13877

Defective homologous recombination DNA repair as therapeutic target in advanced-stage chordoma

We thank the Reviewers and the Associate Editor for their insightful and constructive comments, which have substantially improved this work. Please find enclosed a revised manuscript that has been modified in accordance with their recommendations. Our specific responses to the Reviewers' comments are detailed individually below.

Reviewer #1

This is an interesting paper describing possible homologous recombination DNA repair defects in chordoma, reporting an interesting case with acquired PARP inhibitor resistance and a PARP1 mutation.

We are grateful for these favorable comments, and are delighted that the Reviewer found this an interesting paper.

Major points:

Heterozygous deletion of *ATR*, *BRCA2*, *CHEK2*, *ERCC6*, *FANCA*, *FANCC*, *FANCD2*, *FANCG*, *PALB2*, *RAD18*, *RAD51*, *RAD54L*, or *XRCC3* will not lead to homologous recombination defect. AC3 signature or HRD score may not be an absolute marker of homologous recombination defect. The role of *PTEN* in homologous recombination is controversial (Fraser M, et al. *Clin Cancer Res*. 2012). Therefore, I am not convinced that the chordoma cases other than Chord_03 and Chord_06 are homologous recombination defective cases.

We completely agree that neither mutational signature Alexandrov-COSMIC 3 (AC3) nor a high homologous recombination deficiency (HRD) score alone are sufficient to capture the entire spectrum of HR-deficient tumors. For precisely this reason, we applied an integrated approach that combines multiple markers of defective HR, i.e. (i) germline and somatic alterations of individual HR genes, (ii) specific mutational signatures, (iii) HRD score, and (iv) large-scale state transitions. A growing number of studies have shown that many cases of HR deficiency cannot be explained by mutations in HR genes such as *BRCA1/2*, *ATM*, *PALB2*, *CHEK2*, *NBN*, and others, and that the proportion of cancer patients who may respond to PARP inhibitors is not limited to such cases (e.g. PMID 26632267, 27717299, 28288110, 28851423, 29321523; reviewed in PMID 26775620, 27317574).

As a consequence, there is a need to identify tumors with "BRCAness", i.e. functional defects similar to those associated with *BRCA1/2* inactivation, and we feel that our study makes an important contribution to these ongoing efforts. For example, our genetic and clinical data provide strong evidence that Chord_05, which did not meet

“classical” criteria for defective HR such as presence of an inactivating BRCA1/2 mutation, is a HR-deficient case. This patient showed an exceptionally high exposure to signature AC3, a high degree of genomic instability, and 13 HR gene alterations, although we acknowledge that it is difficult to determine whether these alterations are the cause or the consequence of HR deficiency. In addition, and perhaps most importantly, this patient’s tumor responded to single-agent olaparib for 10 months, and repeat sequencing showed that acquired resistance was associated with a newly gained mutation in PARP1 itself that is predicted to restore enzymatic activity in the presence of drug, thereby validating PARP1 as therapeutic target in this tumor. Bearing in mind that no consensus definition of HR deficiency exists, we would argue that at least four of the seven cases described in the original manuscript are HR-defective (Chord_01, biallelic *CHEK2* inactivation; Chord_03, biallelic *BRCA2* inactivation; Chord_05, see above; Chord_06, biallelic *NBN* inactivation). Since submission of our report, we have analyzed four additional cases, which also show genomic imprints of HR deficiency. These data have been included in the revised manuscript.

We are aware of the report by Fraser et al. (PMID 22114138), who observed that PTEN status was not associated with RAD51 mRNA or protein expression in cultured prostate cancer cells. In contrast, a number of studies support the discovery by Alan Ashworth and colleagues that PTEN has a nuclear function whose disruption causes a HR defect in human tumor cells that confers sensitivity to PARP inhibition both in vitro and in vivo (e.g. PMID 20049735, 20530668, 20944090, 24553445, 25576921, 27741411, 28967905). Based on this evidence, we hope the Reviewer will agree that it was legitimate to include PTEN in an exploratory study that aimed to improve the identification of patients who might benefit from PARP inhibition, which is certainly a useful therapeutic strategy for a wider range of cancers bearing deficiencies in the HR pathway other than just BRCA1/2 mutations.

In our view, the question of whether heterozygous loss of multiple HR genes leads to, or is reflective of, HR deficiency is currently unanswered. Thus, it is difficult, for example, to determine if the striking exposure to signature AC3 (which has been proposed as a novel readout of HR deficiency), high degree of genomic instability, and sensitivity to olaparib treatment observed in patient Chord_05 are related to monoallelic loss of 13 HR genes or another yet unrecognized driver mutation. Of note, there is evidence of dosage effects, including haploinsufficiency, for multiple HR genes (e.g. PMID 10192382, 15282542, 19440510, 21901111), including a recent analysis by the Germline Working Group of the ICGC/TCGA Pan-Cancer Analysis of Whole Genomes Network (doi: <https://doi.org/10.1101/208330>), and clinical studies have identified patients in whom response to PARP inhibition was associated with heterozygous HR gene alterations alone (e.g. PMID 26510020). However, a causal link between multiple HR gene haploinsufficiency and functional impairment of the HR pathway has not yet been established. We have amended the manuscript to reflect these important considerations and, in particular, to emphasize that the definition of HR and the delineation of causative factors are rapidly evolving fields.

The sentence in the abstract “ heavily pretreated chordomas (n=7) are invariably characterized by alterations affecting the homologous recombination (HR) DNA repair pathway...” is an overinterpretation and should be rewritten.

We agree and have reworked this sentence. Our case series, which has now been expanded to include 11 patients, shows that genomic imprints of defective HR are a recurrent feature of advanced chordoma. However, additional studies are needed to determine their precise frequency.

Similarly, the first sentence in Discussion “Our data indicate that a substantial portion of advanced-stage chordomas harbor molecular alterations that affect the repair of DNA damage via HR...” is also incorrect and should be rewritten.

We realize that the term “substantial” was imprecise. In the revised manuscript, we state that our study has uncovered a recurrent pattern of genomic alterations in patients with advanced chordoma. We also appreciate the Reviewer’s point that in several cases, it is difficult to determine whether the alterations detected by whole-exome or genome sequencing are the cause or the consequence of HR deficiency. We therefore chose a more neutral wording, even though patients Chord_03, Chord_06, and Chord_01 did harbor alterations with clear (Chord_03, biallelic *BRCA2* inactivation; Chord_06, biallelic *NBN* inactivation) or likely (Chord_01, biallelic *CHEK2* inactivation) impact on HR function in cancers. Specifically, we now describe that advanced chordomas show “genomic imprints of defective HR repair of DNA double-strand breaks”.

Minor points:

Supplemental Table 1. Age and gender of the patients should be included.

We have included patient age and gender in Supplementary Table 1.

For cases with germline mutations in *BRCA2* or *NBN*, presence or absence of other tumors should be described.

We have included this information in the Results section of the revised manuscript. Both patients had no other tumors.

Reviewer #2

The authors report on WES on 7 normal tumor pairs from patients with pretreated advanced chordoma and report a stabilization of disease in a patient treated with olaparib. These findings are novel and the identification of HR deficiency as a potential target in this rare cancer type is of great interest. Furthermore, the development of a somatic PARP1 mutation in the patient treated with olaparib is quite interesting.

We thank the Reviewer for these positive and encouraging comments.

However, some additional information is needed to interpret these results and the presentation of data seems overly simplistic. Are HR genes significantly over-represented relative to alterations in other genes? No formal analysis is presented to that effect. The authors include a whole list of genes, not all of which have clear impact on HR function in cancers. For example, the role of PTEN and some other genes on the list have not been clearly shown to drive an HRD phenotype either functionally or by mutational signature profiling. The results and discussion should be more nuanced in this regard.

We agree that the Results and Discussion sections were in part not precise enough. In the revised manuscript, we have detailed the criteria underlying our selection of candidate HR genes. Specifically, we selected 12 genes that were assessed as a biomarker to stratify patients with castration-resistant prostate cancer for olaparib treatment in a recent phase 2 clinical trial (PMID 26510020) and 11 additional genes reported to be involved in DNA damage repair or sensitivity to PARP inhibition (*ATR*: PMID 25965342, 27708213; *ERCC6*: PMID 25634215, 27374179; *FANCC/D2/G*: PMID 25609062, 26510020, 28993682; *PTEN*: PMID 20049735, 20944090, 21468130, 23239809, 24625059; *RAD18*: PMID 25417706, 26056084; *RAD51B*: PMID 23239809, 24278037, 29465803; *RAD54L*: PMID 16912188, 26056084, 26669450, 28223274; *RPA1*: PMID 16912188, 23239809; *XRCC3*: PMID 17114795, 23512992, 23760496, 25028150, 29465803).

Concerning the role of PTEN, a number of studies support the initial discovery by Alan Ashworth and colleagues of a nuclear function whose disruption causes a HR defect in human tumor cells that confers sensitivity to PARP inhibition both in vitro and in vivo (e.g. PMID 20049735, 20530668, 20944090, 24553445, 25576921, 27741411, 28967905). We are aware of contradictory results, such as those by Fraser et al. (PMID 22114138) who observed that PTEN status was not associated with RAD51 mRNA or protein expression in prostate cancer cells. However, based on the evidence referenced above, we hope the Reviewer will agree that it was legitimate to include PTEN in an exploratory study that aimed to identify patients who might benefit from PARP inhibition beyond standard criteria.

Given the size of our cohort (original manuscript, $n = 7$; revised manuscript, $n = 11$), which is due to the rarity of chordoma (incidence of fewer than one case per million

persons per year) and the prospective nature of the MASTER program, we cannot determine in a statistically meaningful way whether HR genes are more frequently affected by deletions compared to genes in other functional pathways. Therefore, the question of whether these alterations are a result and not a driver of defective HR has to remain unanswered for the time being. Please see below for a detailed discussion of this important issue, which we have also addressed in the revised manuscript.

What is the difference between deletions and LOH. Are the deletions meant to be homozygous deletions. If not, then how do they differentiate deletions from LOH at that locus? The many deletions in HR gene if only heterozygous and not homozygous are of unclear importance, particularly if they are just passengers in larger chromosomal deletions. Therefore, reporting these “deletions” may be confusing to the reader unless they are occurring at significantly higher frequency in HR genes compared to genes in other functional pathways.

We used the term “deletion” to describe unbalanced genomic losses, whereas “LOH” indicates copy-neutral loss of heterozygosity. As described in the Results section, all deletions were heterozygous. To illustrate this more clearly, we have modified Figure 1c, which had been prepared according to OncoPrint conventions, such that each box is now divided in two parts representing both alleles of the respective gene. In our view, the question of whether heterozygous loss of multiple HR genes leads to, or is reflective of, HR deficiency is currently unanswered. Thus, it is difficult, for example, to determine if the striking exposure to signature AC3 (which has been proposed as a novel readout of HR deficiency), high degree of genomic instability, and sensitivity to olaparib treatment observed in patient Chord_05 are related to monoallelic loss of 13 HR genes or another yet unrecognized driver mutation. Of note, there is evidence of dosage effects, including haploinsufficiency, for multiple HR genes (e.g. PMID 10192382, 15282542, 19440510, 21901111), including a recent analysis by the Germline Working Group of the ICGC/TCGA Pan-Cancer Analysis of Whole Genomes Network (doi: <https://doi.org/10.1101/208330>), and clinical studies have identified patients in whom response to PARP inhibition was associated with heterozygous HR gene alterations alone (e.g. PMID 26510020). However, as alluded to by the Reviewer, a causal link between multiple HR gene haploinsufficiency and functional impairment of the HR pathway has not yet been established. We have amended the manuscript to reflect these important considerations and to emphasize that the definition of HR and the delineation of causative factors are rapidly evolving fields.

The driver of the chord_05 “brca-ness signature” is unclear, but the authors imply that the exceptionally high AC3 signature is also related to the 13 HR gene alterations. A more likely explanation is the frequent LOH or heterozygous deletion of these 13 HR genes is a result of HRD not a driver of HRD, unless one of these events is biallelic. The authors should be clear that the signature of HRD is more important here in choosing a therapy than the many monoallelic deletions of putatively important HR genes. Unless

these monallelic deletions are over-represented in HR genes relative to other genes, their inclusion in Figure 1, panel C is misleading.

We agree and are grateful for these helpful comments. While our genetic and clinical data provide strong evidence that Chord_05 is a HR-deficient case, this tumor did not meet traditional criteria for defective HR, such as biallelic inactivation of *BRCA1/2*. Thus, the underlying driver remained unclear, and our therapeutic choice in this patient was indeed informed by the tumor's overall genomic instability and prominent mutational signature AC3, which highlights the potential of compound genomic measures to identify tumors with deficiencies in the HR pathway that confer sensitivity to PARP inhibition. We also agree with the Reviewer's point that the multiple HR gene losses observed in advanced chordomas may rather be the consequence of defective HR. Of note, there is evidence of dosage effects, including haploinsufficiency, for multiple HR genes (e.g. PMID 10192382, 15282542, 19440510, 21901111), including a recent analysis by the Germline Working Group of the ICGC/TCGA Pan-Cancer Analysis of Whole Genomes Network (doi: <https://doi.org/10.1101/208330>), and clinical studies have identified patients in whom response to PARP inhibition was associated with heterozygous HR gene alterations alone (e.g. PMID 26510020). However, as a causal link between multiple HR gene haploinsufficiency and functional impairment of the HR pathway has not yet been established, the question of whether heterozygous deletion of multiple HR genes can impair HR function remains unanswered. We have amended the manuscript to reflect these important considerations.

I don't think the use of the term BRCAness is helpful without a clear definition of what that means. For example, in the discussion, they reference the "high prevalence of BRCAness in heavily treated chordomas..." What do the authors mean by "BRCAness". This is a poorly defined term. They would be better off describing that 71% had a genomic mutational signature consistent with HRD or a clear biallelic mutation in an HR gene. As stated it is vague and confusing.

We agree and thank the Reviewer for this helpful comment. By using the term "BRCAness", we meant to indicate that many tumors bear deficiencies in the HR pathway that cannot be explained by traditional criteria such as biallelic *BRCA1/2* mutations, and that, as a consequence, a compound measure of defective HR will be needed to identify cancer patients who may respond to PARP inhibitors. However, we realize that in the absence of a consensus definition of "BRCAness", the term lacks precision and should be avoided. In the revised manuscript, we instead specify the proportion of tumors that showed (i) biallelic inactivation of a HR gene, (ii) a mutational signature indicative of defective HR, (iii) an elevated HR deficiency score, or (iv) high numbers of large-scale state transitions, in accordance with the Reviewer's recommendation.

The authors report methods for somatic variant calling but no methods for germline analyses.

Thank you for pointing out this oversight. We have included the methods for germline variant calling.

Furthermore, Figure 1 which summarizes the mutation data is not well annotated in the legend.

We thank the Reviewer for this helpful comment, and have amended the legends to Figure 1a-e to describe the genomic profiles in more detail.

“Recurrent somatic mutations” implies the same mutation is seen across cases. I think the authors mean to say recurrent somatically mutated “genes”

We have reworded this sentence as suggested to avoid any ambiguity.

Reviewer #3

The goal of this study was to identify genomic alterations that can be directly linked to targeted therapy in a rare cancer that shows high frequency of local recurrence and does not have too many treatment options other than surgery. The manuscript was nicely written. However, the study is really small (based on 7 WES) and there are some major problems in study design, data interpretation and conclusion.

We appreciate the Reviewer's feedback on several aspects of our work. Concerning the number of cases studied, we agree but wish to underscore that, with an incidence of fewer than one case per million persons per year, chordoma is an ultra-rare disease. We are not aware of another program that has studied a comparable series of chordoma patients, which has now been expanded to include 11 cases, by whole-exome or genome sequencing in a prospective clinical setting. Thus, while our cohort is inevitably small, it has nevertheless enabled the identification of a recurrent pattern of genomic alterations whose clinical actionability is backed by clinical data, which in turn have led to the discovery of a novel mechanism underlying acquired resistance to pharmacologic PARP inhibition. In our view, this study reinforces that, in the era of comprehensive genomic characterization of individual cancers, small patient cohorts or even " $n = 1$ trials", such as the intervention in patient Chord_05, can be highly informative and guide future research as well as clinical management.

1. With 7 WES, there is no power to identify driver events in these tumors.

We completely agree. We did not aim to map the genomic landscape of chordoma and, therefore, made no attempt to apply computational algorithms for identifying driver genes based on their patterns of mutation in large patient cohorts. As pointed out correctly by the Reviewer in his/her general remarks above, we used whole-exome or genome sequencing in a prospective clinical program to identify therapeutically tractable molecular lesions in patients with an ultra-rare cancer for which no effective medical therapy exists. This aim is clearly stated in the Introduction (last paragraph) and Results (first paragraph) sections of the revised manuscript. As mentioned above, our cohort has meanwhile been expanded and now includes 11 cases.

2. The major conclusion of the impaired HR pathway is not convincing. Most of the alterations are copy number variants (CNVs), however, as shown in the CNV profile figures, most of these CNVs involved large genomic segments containing a huge number of genes. In fact, I did not see any of the listed HR genes showing focal deletions/amplifications.

To identify HR-deficient tumors, we applied an integrated approach that combines multiple markers of defective HR, i.e. (i) germline and somatic alterations of individual HR genes; (ii) a specific mutational signature, Alexandrov-COSMIC 3 (AC3), known to be associated with HR deficiency; (iii) the HR deficiency score; and (iv) the presence of

large-scale state transitions. Thus, our conclusion that genomic imprints of defective HR are a recurrent feature of advanced chordoma was not solely based on the presence of DNA copy number alterations – which met published criteria for HR deficiency (PMID 22933060, 23047548) – but a compound genomic measure. For example, patients Chord_03, Chord_06, and Chord_01 harbor alterations with clear (Chord_03, heterozygous germline *BRCA2* frameshift mutation [ACMG Class 5] accompanied by somatic deletion of the wildtype allele; Chord_06, heterozygous germline *NBN* frameshift mutation [ACMG Class 5] accompanied by somatic deletion of the wildtype allele) or likely (Chord_01, heterozygous germline *CHEK2* missense variant [ACMG Class 4] accompanied by somatic deletion of the wildtype allele) impact on HR function in cancers. On the other hand, patient Chord_05, which did not meet traditional criteria for defective HR such as presence of inactivating *BRCA1/2* mutations, showed an exceptionally high exposure to signature AC3, a high degree of genomic instability, and 13 HR gene deletions, although we acknowledge that it is difficult to determine whether these genomic losses are the cause or the consequence of HR deficiency. In addition, this patient's tumor responded to single-agent olaparib for 10 months, and repeat sequencing showed that acquired resistance was associated with a newly gained mutation in *PARP1* itself that is predicted to restore enzymatic activity in the presence of drug, thereby validating *PARP1* as therapeutic target in this tumor. Signature AC3, which has been proposed as a novel readout of HR deficiency, contributed to the mutational catalog in all tumors, and the 95% confidence interval of the exposure to AC3 did not contain zero in more than 70% of samples. Comparison of the signatures identified in our patients against a background of 7,042 cancer samples demonstrated significant enrichment of AC3. Based on these considerations, we hope the Reviewer will agree that our collective genetic and clinical data provide strong evidence for genomic imprints of defective HR as recurrent feature of advanced chordoma. In addition, we have analyzed four new cases since the submission of our report, which all show a genomic profile consistent with HR deficiency. These data have been included in the revised manuscript.

3. In most cases, genes in a single pathways display mutations in a mutually exclusive manner, rather than co-occurrence, which usually reflects non-significant mutations.

In our view, it is not generally true that the co-occurrence of mutations in the same pathway argues against their functional relevance. Important examples include the association of (i) non-V600 *BRAF* mutations with activating mutations in receptor tyrosine kinases or *RAS* family members; (ii) *PIK3CA* mutations with alterations of various driver genes such as *PTEN*, *EGFR*, *ALK*, *KRAS*, *BRAF*, and *MEK1*; and (iii) *GATA2* mutations with alterations of other myeloid transcription factors such as *CEBPA* and *EVI1*. In the case of alterations affecting genes involved in DNA repair via HR, we acknowledge that it is difficult to discriminate between alterations that clearly drive HR deficiency, such as biallelic inactivation of *BRCA2* in patient Chord_03, and alterations that may rather be the consequence of HR deficiency, such as the multiple genomic losses observed in patient Chord_05. We have modified the Discussion section of the manuscript to highlight this challenge.

4. All these patients were treated with either RT or chemo before tumor specimens were obtained. The genomic profile may be related to treatment.

As discussed in the manuscript (Discussion section, second paragraph), we agree with the Reviewer that the tumors' genomic profiles may have been shaped by prior exposure to ionizing radiation. This scenario is particularly interesting from a clinical perspective because it raises the possibility that standard treatment of chordoma may induce a targetable vulnerability. On the other hand, previous microarray analyses that included radiation-naïve cases indicate that genomic instability may be an intrinsic feature of chordoma (e.g. PMID 21602918). Given that precision oncology programs such as ours typically enroll patients who have exhausted standard therapies, we cannot discriminate between treatment-related and disease-specific genomic changes based on the cases that we have studied by whole-exome or genome sequencing. As alluded to in the Discussion section of the manuscript, we believe that the imprint of ionizing radiation in cancer is an important yet understudied issue that should be investigated in future studies.

In contrast, we consider it unlikely that the genomic changes observed in our patients were induced by medical therapy as advanced chordoma is usually treated with agents targeting kinase signaling pathways (Chord_03, Chord_05, and Chord_11, imatinib; Chord_06, imatinib and sirolimus; Chord_10, imatinib, sunitinib, everolimus, erlotinib, and bevacizumab) and not DNA-damaging chemotherapy, with dedifferentiated chordoma being a notable exception (Chord_07, doxorubicin and ifosfamide).

Reviewers' comments:

Reviewer #1 (Remarks to the Author):

The authors have addressed all of the concerns I raised.

Minor points:

1) Supplemental Table 6 is missing.

Reviewer #2 (Remarks to the Author):

This revision is much improved. However, there is still one major issue which the authors persist on which is going to mislead readers. This is regarding the presence of het deletions in HR genes which are non-focal and are just a consequence and not a cause of HRD as perhaps explaining the response to PARPi. It is better to say that the reason for HRD in this case is not known. In the response letter the authors state that, "due to the size of the cohort, we cannot determine in a statistically meaningful way whether HR genes are more frequently affected by deletions compared to genes in other functional pathways". This response is not accurate or adequate. That's type of analysis can be done with WES data on a case by case basis. For each case, the authors can determine the ontology of genes with het deletions or LOH versus those without. They can then do a statistical analysis to determine if certain pathways are over-represented in the deletions (z score) for that one case. then repeat for each case. If these deletion events are not over-represented in HR genes they are almost certainly a consequence, not a cause of HRD. This analysis is not that complicated and should be done. If the data do not support a causal role for the deletions, then they should stop talking about specific deletion events, and they should not be linking the presence of these deletions to the cause of HRD. The authors attempt to be more nuanced in this regard, but have just muddied the water further in the discussion:

"Of note, most patients in our study also exhibited monoallelic HR gene deletions. However, the question of whether heterozygous loss of multiple HR genes leads to, or is reflective of, HR deficiency is currently unanswered." TRY TO ANSWER IT! "Thus, it is difficult to determine, e.g., if the genomic features and sensitivity to olaparib

treatment observed in patient Chord_05 were related to monoallelic loss of 13 HR genes or another yet

unrecognized driver mutation. There is evidence of dosage effects, including haploinsufficiency, for multiple HR

genes^{26–30}, and a recent clinical study has identified patients in whom response to PARP inhibition was associated with heterozygous HR gene alterations alone¹⁵." this whole discussion has nothing to do with widespread LOH affecting many large chromosomal segments and does not support their specific cases.

I will say again, if they are going to make this claim, then they must attempt to answer the question. If this is a cause and not a consequence of HRD, then the HR genes will be statistically over-represented in these events.

Unless the correct analysis really proves a role for HR genes events being significant, then they should just mention that many genes have LOH events including many HR genes and many non DNA repair genes and discuss those events that are more specific than that (specific mutations and biallelic events as they do mention). It is misleading to list all the HR deletions in the results of each tumor without listing all the rest of the other genes with deletions. They are cherry picking the results they think fit with the hypothesis. It is this type of thinking that leads to precision medicine reports suggesting PARPi for every LOH event occurring in a HR gene for each cancer, which is not accurate. In this case, it was the mutational signature and HRD signature that is the key to choosing the drug.

Reviewer #3 (Remarks to the Author):

Although the responses from the authors are very helpful, I still have the problem with the interpretation based on the genomic data, particularly those frequencies of HR altered tumors presented in the second paragraph of Results based on Fig 1c. In the recently published Chordoma genomic landscape study which is based on a much larger sample size, cited as ref10 in the current manuscript, HR genes were not identified as driver or even recurrent mechanisms.

I agree that Chordoma is very rare and clinical findings based on small studies can be very informative. However, the conclusion based on genomic data is not convincing. Can you reframe it as a clinical case report?

RE: NCOMMS-18-13877A

Defective homologous recombination DNA repair as therapeutic target in advanced chordoma

We thank the Reviewers and the Associate Editor for their insightful and constructive comments, which have further improved this work. Please find enclosed a revised manuscript that has been modified in accordance with their recommendations. Our specific responses to the Reviewers' comments are detailed individually below.

Reviewer #1

The authors have addressed all of the concerns I raised.

Minor points:

1) Supplemental Table 6 is missing.

We apologize for this oversight and have included Supplementary Table 6.

Reviewer #2

This revision is much improved.

Thank you.

However, there is still one major issue which the authors persist on which is going to mislead readers. This is regarding the presence of het deletions in HR genes which are non-focal and are just a consequence and not a cause of HRD as perhaps explaining the response to PARPi. it is better to say that the reason for HRD in this case is not known. In the response letter the authors state that, "due to the size of the cohort, we cannot determine in a statistically meaningful way whether HR genes are more frequently affected by deletions compared to genes in other functional pathways". This response is not accurate or adequate. Thats type of analysis can be done with WES data on a case by case basis. For each case, the authors can determine the ontology of genes with het deletions or LOH versus those without. They can then do a statistical analysis to determine if certain pathways are over-represented in the deletions (z score) for that one case. then repeat for each case. If these deletion events are not ever-represented in HR genes they are almost certainly a consequence, not a cause of HRD. This analysis is not that complicated and should be done. if the data do not support a causal role for the deletions, then they should stop talking about specific deletion events, and they should not be linking the presence of these deletions to the cause of HRD. the authors attempt to be more nuanced in this regard, but have just muddied the water further in the discussion: "Of note, most patients in our study also exhibited monoallelic HR gene deletions. However, the question of whether heterozygous loss of multiple HR genes leads to, or is reflective of, HR deficiency is currently unanswered." TRY TO ANSWER IT! "Thus, it is difficult to determine, e.g., if the genomic features and sensitivity to olaparib treatment observed in patient Chord_05 were related to monoallelic loss of 13 HR genes or another yet unrecognized driver mutation. There is evidence of dosage effects, including haploinsufficiency, for multiple HR genes^{26–30}, and a recent clinical study has identified patients in whom response to PARP inhibition was associated

with heterozygous HR gene alterations alone¹⁵." this whole discussion has nothing to do with widespread LOH affecting many large chromosomal segments and does not support their specific cases. I will say again, if they are going to make this claim, then they must attempt to answer the question. If this is a cause and not a consequence of HRD, then the HR genes will be statistically over-represented in these events.

As recommended by the Reviewer, we have performed a statistical analysis to evaluate if homologous recombination (HR) genes are significantly overrepresented among the genes affected by heterozygous deletion. For each patient, we determined (i) the number of deleted and non-deleted protein-coding genes (based on GENCODE 19) and (ii) the number of deleted and non-deleted HR genes (based on the 23 loci given in Supplementary Table 5). The resulting counts were arranged as 2 x 2 contingency tables, and for each patient a one-sided Fisher exact test was used to assess statistical significance.

Cases analyzed by whole-exome sequencing:

Chord_01		HR status	
		HR genes	Non-HR genes
Deletion status	Deleted genes	46	4959
	Non-deleted genes	148	15089
P value		0.408	

Chord_02		HR status	
		HR genes	Non-HR genes
Deletion status	Deleted genes	9	840
	Non-deleted genes	185	19208
P value		0.702	

Chord_03		HR status	
		HR genes	Non-HR genes
Deletion status	Deleted genes	50	5231
	Non-deleted genes	144	14817
P value		0.498	

Chord_04		HR status	
		HR genes	Non-HR genes
Deletion status	Deleted genes	20	2650
	Non-deleted genes	174	17398
P value		0.138	

Chord_05		HR status	
		HR genes	Non-HR genes
Deletion status	Deleted genes	61	6942
	Non-deleted genes	133	13106
P value		0.198	

Chord_06		HR status	
		HR genes	Non-HR genes
Deletion status	Deleted genes	63	6484
	Non-deleted genes	131	13564
P value		0.550	

Chord_07		HR status	
		HR genes	Non-HR genes
Deletion status	Deleted genes	65	6989
	Non-deleted genes	129	13059
P value		0.387	

Chord_08		HR status	
		HR genes	Non-HR genes
Deletion status	Deleted genes	40	3807
	Non-deleted genes	154	16241
P value		0.751	

Chord_09		HR status	
		HR genes	Non-HR genes
Deletion status	Deleted genes	40	4219
	Non-deleted genes	154	15829
P value		0.484	

Cases analyzed by whole-genome sequencing:

Chord_10		HR status	
		HR genes	Non-HR genes
Deletion status	Deleted genes	41	4682
	Non-deleted genes	153	15366
P value		0.263	

Chord_11		HR status	
		HR genes	Non-HR genes
Deletion status	Deleted genes	26	2896
	Non-deleted genes	168	17152
P value		0.387	

We also performed an analysis across the entire cohort by assigning a gene to the deleted group if it showed a genomic loss in any of the samples:

Cohort		HR status	
		HR genes	Non-HR genes
Deletion status	Deleted genes	122	12661
	Non-deleted genes	72	7387
P value		0.4967	

Unless the correct analysis really proves a role for HR genes events being significant, then they should just mention that many genes have LOH events including many HR genes and many non DNA repair genes and discuss those events that are more specific than that (specific mutations and biallelic events as they do mention). It is misleading to list all the HR deletions in the results of each tumor without listing all the rest of the other genes with deletions. They are cherry picking the results they think fit with the hypothesis. It is this type of thinking that leads to precision medicine reports suggesting PARPi for every LOH event occurring in a HR gene for each cancer, which is not accurate. In this case, it was the mutational signature and HRD signature that is the key to choosing the drug.

We agree and have addressed this important issue in several ways. Based on the results of the above statistical analyses, which indicate that HR genes are not overrepresented among the many loci affected by deletion and/or loss of heterozygosity (LOH) in chordoma genomes, we have modified both the presentation of results and our conclusions as suggested by the Reviewer. In particular, we have omitted the description of specific heterozygous deletions or LOH events from the Results section and Figure 1c, unless they had occurred as “second hit” resulting in complete inactivation of the respective HR gene. Accordingly, we now place more emphasis on specific mutations and cases with biallelic events. Furthermore, we discuss that in several cases, including Chord_05, the driver of the HR deficiency “footprint” (defined by mutational signature Alexandrov-COSMIC 3 [AC3], the HR deficiency score, and the presence of large-scale state transitions) is unknown. Finally, we stress that our therapeutic choice in patient Chord_05 was primarily informed by the tumor’s overall genomic instability and prominent signature AC3, which highlights the potential of compound genomic measures to identify HR-deficient tumors that do not meet traditional criteria for defective HR, such as biallelic inactivation of *BRCA1/2*.

Reviewer #3

Although the responses from the authors are very helpful, I still have the problem with the interpretation based on the genomic data, particularly those frequencies of HR altered tumors presented in the second paragraph of Results based on Fig 1c.

We are delighted that the Reviewer found our responses to the prior critiques helpful. As discussed in our response to Reviewer #2, we agree that the presentation of specific heterozygous deletions and LOH events was misleading, and we have modified the Results section and Figure 1c accordingly. Please see above for details.

In the recently published Chordoma genomic landscape study which is based on a much larger sample size, cited as ref10 in the current manuscript, HR genes were not identified as driver or even recurrent mechanisms.

To identify HR-deficient tumors, we applied an integrated approach that combines multiple markers of defective HR, i.e. (i) germline and somatic alterations of individual HR genes; (ii) mutational signature AC3, known to be associated with HR deficiency; (iii) the HR deficiency score; and (iv) the presence of large-scale state transitions. Importantly, the majority of these markers were not investigated by Tarpey et al. For example, the study included no in-depth analyses of genome-wide DNA copy number alterations and mutational signatures and no evaluation of germline alterations. In addition, no treatment data were reported by Tarpey et al. Thus, it is not known if the tumors analyzed had been subjected to radiotherapy, which may have shaped their genomic profiles as alluded to by Reviewer #3 previously, and if any patients had received therapies that are known to be effective in HR-deficient cancers, such as platinum derivatives or PARP inhibitors. Based on these considerations, we hope the Reviewer will agree that the report by Tarpey et al. allows no conclusion as to the occurrence of HR deficiency in patients with advanced-stage chordoma and, thus, does not mitigate against our data. In fact, we believe that the work by Tarpey and colleagues and our study are quite complementary, as the former represents a typical “genomic landscape” study that primarily focused on the identification of individual chordoma driver genes affected by single-nucleotide variants and small insertions and deletions, whereas our report is centered on a complex, yet clinically “actionable”, molecular profile that was discovered using a compound genomic measure.

I agree that Chordoma is very rare and clinical findings based on small studies can be very informative. However, the conclusion based on genomic data is not convincing. Can you reframe it as a clinical case report?

While Chord_05 certainly is the most informative case, we hope the Reviewer will agree that a lot of critical information would be lost if the report was limited to this patient. For example, patients Chord_03, Chord_06, and Chord_01 harbored alterations with clear (Chord_03, heterozygous germline *BRCA2* frameshift mutation [ACMG Class 5] accompanied by somatic deletion of the wildtype allele; Chord_06, heterozygous germline *NBN* frameshift mutation [ACMG Class 5] accompanied by somatic deletion of the wildtype allele) or likely (Chord_01, heterozygous germline *CHEK2* missense variant [ACMG Class 4] accompanied by somatic deletion of the wildtype allele) impact on HR function in cancers. Furthermore, mutational signature AC3 was significantly enriched in 72.7% of samples and coincided with HR deficiency-related patterns of genomic instability, i.e. elevated HR deficiency scores and high numbers of large-scale state transitions. Thus, our conclusion that genomic imprints of defective HR are a

recurrent feature of advanced chordoma is clearly supported by the collective genetic and clinical data from the entire patient cohort, not just Chord_05.

REVIEWERS' COMMENTS:

Reviewer #3 (Remarks to the Author):

The major points raised previously by reviewers 2 and 3 were addressed. The authors took out the section of large CNVs involving HR genes as a major driver mechanism.

However, the conclusion that chordomas are frequently characterized by genomic patterns indicative of defective homologous recombination (HR) DNA repair and alterations affecting HR-related genes is still an overstatement based on germline change and mutation signature. The presence of germline variants does not indicate disease causality. Given that only a small number of genes were evaluated, it is unknown whether and how pathogenic variants in other genes contribute to disease predisposition. Due to the small number of patients, it is impossible to test whether the genetic burden associated with these HR genes is indeed higher in Chordoma patients than in controls. The estimates of mutation signatures primarily based on exome sequencing in such a small number of tumors with very low mutation burden may not be accurate. Moreover, it is not clear whether the increased HRD is a cause of the disease or consequence of the treatment.

RE: NCOMMS-18-13877B
Defective homologous recombination DNA repair as therapeutic target in advanced chordoma

Reviewer #3

The major points raised previously by reviewers 2 and 3 were addressed. The authors took out the section of large CNVs involving HR genes as a major driver mechanism. However, the conclusion that chordomas are frequently characterized by genomic patterns indicative of defective homologous recombination (HR) DNA repair and alterations affecting HR-related genes is still an overstatement based on germline change and mutation signature.

As described in our manuscript and discussed in previous rounds of review, we identified multiple genomic features that are widely known to be associated with defective HR:

- 27.3% of patients harbored pathogenic (according to ACMG criteria) germline alterations of established HR genes that were accompanied by somatic deletion of the respective wildtype alleles. Thus, our conclusions are not based on the detection of germline changes alone. Biallelic inactivation of, e.g., *BRCA2* is known to be associated with HR deficiency.
- 72.7% of samples showed significant enrichment of mutational signature AC3, which is known to be associated with HR deficiency.
- Enrichment of signature AC3 coincided with elevated HR deficiency scores and high numbers of large-scale state transitions. Thus, our conclusions are not based on alterations of individual HR genes and mutational signatures alone, but also take into account specific patterns of genomic instability. Elevated HR deficiency scores and high numbers of large-scale state transitions are known to be associated with HR deficiency.

The presence of germline variants does not indicate disease causality. Given that only a small number of genes were evaluated, it is unknown whether and how pathogenic variants in other genes contribute to disease predisposition.

As discussed in previous rounds of review, we did not study disease causality or predisposition, and our manuscript makes no claims in this regard. We report the discovery of a clinically actionable genetic profile in patients with advanced-stage chordoma.

Due to the small number of patients, it is impossible to test whether the genetic burden associated with these HR genes is indeed higher in Chordoma patients than in controls.

We do not understand what the Reviewer means by “genetic burden associated with these HR genes”. We did not study tumor mutational burden but focused on qualitatively distinct genomic features that are specifically associated with HR deficiency.

The estimates of mutation signatures primarily based on exome sequencing in such a small number of tumors with very low mutation burden may not be accurate.

We disagree with this point, which did not arise in previous rounds of review. We analyzed two cases by WGS and nine cases by WES. The SNV load of the WGS cases exceeded any known threshold for the feasibility of mutational signature analysis (2,929 and 2,048 SNVs, respectively). Likewise, the total SNV burden of the WES samples was not very low but ranged from 35 to 293 (mean, 97; median, 66). As detailed in the Methods section, we used the R package YAPSA to perform supervised analysis of mutational signatures, which requires much less statistical power of the input data than unsupervised analysis. This software has been used to analyze SNV calls from thousands of cancer samples (e.g. PMID 28474103, 28572216, 29321523, 29748622, 30537516, López et al., accepted for publication in *Nature Communications*), and it has emerged that mutational signatures can be determined if the SNV load of a sample is above 25. All chordoma samples analyzed in our study had mutational loads above this threshold. The detection of a common subset of mutational signatures in this entity, both among the different WES samples and between WES and WGS samples, underscores the stability of the calling algorithm.

Moreover, it is not clear whether the increased HRD is a cause of the disease or consequence of the treatment.

These points are already addressed in the manuscript and were extensively discussed in previous rounds of review.